



# Quantification of Uncertainties in OCO-2 Measurements of XCO2: Simulations and Linear Error Analysis

Brian Connor[1], Hartmut Bösch[2,6], James McDuffie[3], Tommy Taylor[4], Dejian Fu[3], Christian Frankenberg[5], Chris O'Dell[4], Vivienne H. Payne[3], Michael Gunson[3], Randy Pollock[3], Jonathan Hobbs[3], Fabiano Oyafuso[3], and Yibo Jiang[3]

[1] BC Scientific Consulting, Stony Brook, NY, USA
[2] EOS Group, Department of Physics and Astronomy, University of Leicester, Leicester, UK
[3] Jet Propulsion Laboratory, California Institute of Technology, Pasadena, CA, USA
[4] Cooperative Institute for Research in the Atmosphere, Fort Collins, CO, USA
[5] California Institute of Technology, Pasadena, CA, USA
[6] National Centre for Earth Observation NCEO, University of Leicester, Leicester, UK

*Correspondence to*: Brian Connor (bc.scientific.consulting@gmail.com)

**Abstract.** We present an analysis of uncertainties in global measurements of the column averaged dry-air mole fraction of $CO_2$ ('$XCO_2$') by the NASA Orbiting Carbon Observatory-2, ('OCO-2'). The analysis is based on our best estimates for uncertainties in the OCO-2 operational algorithm and its inputs, and uses simulated spectra calculated for the actual flight and sounding geometry, with measured

atmospheric analyses. The simulations are calculated for land nadir and ocean glint observations. We include errors in measurement, smoothing, interference, and forward model parameters. All types of error are combined to estimate the uncertainty in $XCO_2$ from single soundings, before any attempt at bias correction has been made. From these results we also estimate the 'variable error' which differs between

soundings, to infer the error in the difference of $XCO_2$ between any two soundings. The most important error sources are aerosol interference, spectroscopy, and instrument calibration. Aerosol is the largest source of variable error. Variable errors are usually < 1 ppm over ocean and ~0.5 – 2.0 ppm over land. The total error due to all sources is ~1.5 – 3.5 ppm over land, ~1.5 – 2.5 ppm over ocean.





## 1. Introduction

The Orbiting Carbon Observatory-2 (OCO-2) was launched on July 2, 2014 and has been making global measurements of $CO_2$ and $O_2$ spectral bands in reflected sunlight since early September 2014. Spectra are recorded in two $CO_2$ bands at 1.61 μm and 2.06 μm ('WCO2' and 'SCO2', respectively), and the $O_2$ A-band at 0.76 μm with a resolving power between 17000 and 20000. These measurements are analyzed to

provide estimates of the column-averaged dry-air mole fraction of $CO_2$, known as $XCO_2$. Details of the OCO-2 mission, measurement technique, and $XCO_2$ retrieval may be found in Crisp et al, (2008) and O'Dell et al, (2012). The instrument calibration is detailed in Rosenberg et al, (2016) and Lee et al, (2016). These measurements are motivated primarily by the need to infer regional carbon fluxes, and

to constrain global models of the carbon cycle. Characterizing the uncertainties in $XCO_2$ as measured, and in how these uncertainties vary in space and time, is critical for this purpose. The present study is part of the ongoing effort at uncertainty quantification for the OCO-2 mission.

In this paper we assess the uncertainty in a single $XCO_2$ sounding by 'bottom-up'

analysis, using the best available estimates of errors in the algorithm and its inputs ('error sources'), computing the contribution of each error source to uncertainty in $XCO_2$, and combining them to estimate net uncertainty. We further estimate the uncertainty in the difference of $XCO_2$ between any two soundings, by excluding the mean error produced by error sources which are constant in themselves. We compare

$XCO_2$ uncertainties and their variability over land (nadir) and ocean (glint) for both June and December simulated data sets. Finally, we summarize the effect of all error sources globally, and identify those which are largest and most variable.

Our methodology is described in detail in Sections 2 and 3 below. In overview,

simulations of OCO-2 spectra were run with the CSU orbit simulator (O'Brien et al., 2009), retrievals were performed with the operational Level-2 (L2) code, and a linear error analysis was performed using a dedicated off-line code. This study used simulations to allow full control of the calculations and their inputs. The use of simulations for this analysis should not significantly affect its overall conclusions or

their applicability to OCO-2 operational measurements, since the simulations are



calculated from 'true states' drawn from a realistic atmosphere.

Simulated spectra were calculated for 3 days in June and 3 in December. Only nadir
spectra were calculated over land, and only glint spectra over ocean. Calculations

were done for a single footprint (as opposed to 8 footprints for the flight data), with
the sounding frequency set at 1 Hz (as compared to 3 Hz in operation). The resulting
reduction in data volume (factor of 24) was done purely for convenience and should
not affect conclusions from this study. After cloud screening using the Oxygen A-
Band Preprocessor (ABP) (Taylor, et al, 2016), the operational L2 code was run on

the simulations, and the results screened for convergence. A second screen was
performed to minimize the occurrence of outliers, by restricting the accepted range of
some sounding and retrieved parameters. More than 20 thousand soundings passed
these screens, and those were run with the L2 code a second time, using an extended
state vector including a set of interfering aerosols. The interfering aerosols are tightly

constrained to very small values in this step, but are included to force the L2 code to
calculate their Jacobians. The Jacobian matrix, K, for the extended state vector was
saved for later analysis. (The Jacobian matrix contains the first order partial
derivatives of the forward model with respect to the state vector elements, i.e.,
$K=dF(x)/dx$. In addition, Jacobians were evaluated for a range of forward model

errors, and also saved.

Linear error analysis (Rodgers, 2000, Rodgers & Connor, 2003) was performed on the
extended L2 output, using an off-line code developed for the purpose (Connor et al,
2008). We calculate actual uncertainties using estimates of true error in the measured

spectrum, the variability of the atmospheric ensemble, and forward model errors.
These calculations are intended to apply to a comparison of L2 results to the true
atmospheric values, without applying a 'bias correction' (Wunch, et al, 2011) to the
L2 results.

The paper is organized as follows. In Sect. 2 we briefly discuss the L2 retrieval
algorithm and then present details of the error analysis methodology. This is followed
by an enumeration and discussion of the error sources to be considered in Sect. 3. Sect.
4 contains the results of the linear error analysis. Sect. 5 is a discussion of the results,
and Sect. 6 identifies needs for future research.



## 2. Background and Methodology

The OCO-2 level 2 full physics retrieval algorithm ('L2'), consists of a forward model and inverse method, described in full detail in JPL (2015). The forward model is a radiative transfer model of the atmosphere coupled to a model of the solar spectrum to calculate the monochromatic spectrum at the top of the atmosphere, which is then convolved with the response function as measured for the OCO-2 instrument. The inverse method is a maximum a posteriori likelihood method of a type which has been widely used in the community (Rodgers, 2000; Rodgers & Connor, 2003; Connor et al, 2008, O'Dell et al, 2012). Uncertainty in the OCO-2 measurements of $XCO_2$ has been assessed using an off-line error analysis code developed for the purpose (Connor et al, 2008).

### 2.1 Formulation

The error analysis algorithm performs a linear analysis using Jacobians calculated by the operational OCO-2 forward model. This section closely follows the discussion in Connor et al, (2008).

As defined in JPL (2015), $\mathbf{S_a}$ is the *a priori* covariance matrix, $\mathbf{S_\varepsilon}$ is the measurement error covariance matrix, and $\mathbf{K}$ is the weighting function (Jacobian) matrix. The off-line calculations are more detailed and more realistic than error estimates performed operationally. For example, if forward model errors are included in the $\mathbf{S_\varepsilon}$ matrix used operationally, the retrieved state may be systematically biased by the *a priori* state. Thus we evaluate the effect of forward model errors off-line. Further, evaluation of the smoothing and interference errors strictly requires the covariance of the ensemble of true states, $\mathbf{S_c}$, which is not necessarily equal to the *a priori* covariance $\mathbf{S_a}$ (Rodgers & Connor, 2003). The authors' experience with other remote sensing retrievals suggests that the *a priori* constraint, embodied in $\mathbf{S_a}$, should be as uniform as practical over all soundings, to avoid introducing an additional source of variability. However, the covariance of true states, $\mathbf{S_c}$, varies with latitude, longitude, and season. Estimates of $\mathbf{S_c}$ are readily included in the off-line error estimates. (See for example section 3.3.1.)



Equations 1 through 6 follow the definitions of Rodgers (2000) and Rodgers & Connor (2003). Given $\mathbf{K}$, $\mathbf{S_\varepsilon}$, and $\mathbf{S_a}$, we first characterize the operational retrieval by

calculating the gain function $\mathbf{G_y}$ and the averaging kernel matrix $\mathbf{A}$:

$$\mathbf{G_y} = (\mathbf{K}^T \mathbf{S_\varepsilon}^{-1} \mathbf{K} + \mathbf{S_a}^{-1})^{-1} \mathbf{K}^T \mathbf{S_\varepsilon}^{-1} \tag{1}$$

and

$$\mathbf{A} = \mathbf{G_y} \mathbf{K} \tag{2}$$


We then specify a list of estimated errors to include in the calculation, and where possible the correlation between errors. We will refer to these as 'error sources.' Next we assemble these into an ensemble covariance $\mathbf{S_c}$ (for elements in the state vector) and a forward model parameter covariance $\mathbf{S_b}$ (for elements not included in the state

vector). Finally, we calculate the Jacobian matrix with respect to the forward model parameters, denoted $\mathbf{K_b}$.

For each error in the list, we calculate the resulting covariance of the retrieved state vector, as follows. For measurement error,


$$\hat{\mathbf{S}}_m = \mathbf{G_y} \mathbf{S_\delta} \mathbf{G_y}^T \tag{3}$$

where $\mathbf{S_\delta}$ may be equal to $\mathbf{S_\varepsilon}$, or an alternative estimate of the actual measurement covariance. In the work presented here, $\mathbf{S_\delta} = \mathbf{S_\varepsilon}$.


For forward model error,

$$\hat{\mathbf{S}}_f = \mathbf{G_y} \mathbf{K_b} \mathbf{S_b} \mathbf{K_b}^T \mathbf{G_y}^T \tag{4}$$


For smoothing error,

$$\hat{\mathbf{S}}_s = (\mathbf{A} - \mathbf{I}) \mathbf{S_c} (\mathbf{A} - \mathbf{I})^T \tag{5}$$



where I is the identity matrix.

And for interference error, which refers to error in $CO_2$ caused by non-$CO_2$
components of the state vector

$$\hat{S}_i = A_{ue}\, S_{ec}\, A_{ue}{}^T \qquad (6)$$

where $S_{ec}$ is the ensemble covariance for the non $CO_2$ elements $e$, and $A_{ue}$ is the off-
diagonal block of the averaging kernel matrix which relates $e$ to the $CO_2$ profile $u$.

Finally, the total covariance is

$$\hat{S} = \hat{S}_m + \hat{S}_s + \hat{S}_i + \hat{S}_f \qquad (7)$$

and the resulting variance of $XCO_2$ is $\sigma^2{}_{XCO2} = h^T \hat{S}\, h$, where $h = \partial X_{CO2} / \partial x$

represents the pressure weighting function. Alternatively, one may calculate the
variance in $XCO_2$ due to a given error source, r, as $\sigma_r^2 = h^T\, \hat{S}_r\, h$ and sum the variances
for all r.

The discussion of the preceding paragraph makes two assumptions. One, that the
retrieval is approximately linear within the region bounded by its uncertainty, and
two, that the error sources considered are themselves uncorrelated. Whenever error
sources are correlated, the correlations must be included in, e.g. Eq (4) or (6), and the
net effect on $XCO_2$ calculated for the combined correlated sources.

**2.2 Treatment of Fixed Error Sources**

Many of the error sources we will consider do not vary randomly, and some do not
vary at all.  Spectroscopic errors belong to the class of error sources which are truly
fixed. Unfortunately, due to the varying amount of information in each measured
spectrum relative to the *a priori* constraint, embodied in changes in the gain function,
$G_y$, the resulting errors in retrieved $XCO_2$ are not fixed. We will treat such errors as
follows.



We note that the gain function, $\mathbf{G_y}$, represents the sensitivity of the state vector to the measured radiances. Combining that with the definition of $\mathbf{K_b}$, and considering for the

moment a single scalar parameter, the error caused by parameter $\mathbf{b}$ is

$$\hat{\mathbf{x}} - \mathbf{x} = \mathbf{h^T G_y K_b db} \tag{8}$$

where $\hat{\mathbf{x}} - \mathbf{x}$ is the retrieved $XCO_2$ minus the true $XCO_2$, or we may write


$$\hat{\mathbf{x}} - \mathbf{x} = (\mathbf{h^T G_y K_b db^2 K_b^T G_y^T h})^{1/2} \tag{9}$$

Replacing $\mathbf{db}^2$ with its matrix equivalent $\mathbf{S_b}$, then for an ensemble of retrievals,

$$\sigma_x = \mathbf{rms}(\hat{\mathbf{x}} - \mathbf{x}) = \mathbf{rms}[(\mathbf{h^T G_y K_b S_b K_b^T G_y^T h})^{1/2}] \tag{10}$$

So if $\mathbf{db}$ is a constant, the error $\hat{\mathbf{x}} - \mathbf{x}$ caused by it will vary about a mean value given by $\sigma_x$. While the true error in parameter $\mathbf{b}$ is an unknown constant, we assume that error is equal to the uncertainty in $\mathbf{b}$.


**2.3 Variable Error**

Sources and sinks of $CO_2$ and the circulation of the atmosphere produce temporal and spatial gradients in the $XCO_2$ field, which are quantitatively predicted by carbon cycle models. Measuring these gradients is a strong test of such models. Thus, errors which

vary from sounding to sounding limit the efficacy of the OCO-2 measurements in constraining carbon cycle models. On the other hand, an error which is constant, or at least has a well-defined mean value, can be subtracted from all soundings with minimal or no effect on gradients of $XCO_2$. Therefore, we have attempted to distinguish the uncertainty which differs between soundings, i.e. applies to a

difference in two soundings, from the total accuracy.

The term 'variable error' will be used to refer to a composite error calculated from a selection taken from all error sources described above. Variable error will be



calculated from all error sources, but will exclude the mean error produced by fixed

error sources as discussed in Sect. 2.2. Then a first approximation to the predicted

error in the difference of $XCO_2$ between two soundings will simply be the variable

error multiplied by $\sqrt{2}$, assuming remaining errors are uncorrelated in space or time.

This should be equivalent to estimating the net uncertainty in each sounding, and

assuming a simple bias correction relative to validation observations has been

performed.


## 3. Error Types

We will consider four types of error: measurement, smoothing, interference, and

forward model.


### 3.1 Measurement error

The first and most obvious error is random noise in the measured spectrum. This is

calculated based on the operational noise model (JPL, 2015), and its direct effect on

$XCO_2$ is calculated, and tabulated as 'measurement error.'


However, it is observed that spectral residuals do not decrease with averaging as

would be expected for pure random noise, but instead have a systematic structure.

Because of this it was decided to derive empirical orthogonal functions (EOFs)

representing this systematic structure, and to retrieve scale factors for these functions

at every sounding. (See Section 3.3.2.6 of JPL, (2015)).  Uncertainties in this process

are to be addressed as interference error, below.

### 3.2 Smoothing error

This represents error due to the *a priori* constraint of the state vector. As suggested by

Rodgers & Connor (2003), we have separated this into two components. The first,

smoothing by the true $CO_2$ profile, which we simply refer to as 'smoothing', is

discussed here. The second component is error introduced into $XCO_2$ by the non-$CO_2$



elements of the state vector, which we call 'interference', discussed in the following section.


The error due to the true atmospheric $CO_2$ profile would be best estimated by using the covariance of the ensemble of true states, $\mathbf{S_c}$. Exactly which states to include in the ensemble is not well defined. We have chosen to use the *a priori* covariance $\mathbf{S_a}$, which is intended to represent the variability of $CO_2$ globally throughout the year. We will

systematically overestimate the smoothing error as a result. However, the smoothing error is always small, as we will see, and the use of $\mathbf{S_a}$ is fundamentally conservative.

### 3.3 Interference error

3.3.1 Aerosol and Cloud

We apply the Modern Era Retrospective analysis for Research and Applications (MERRA) aerosol reanalysis climatology for daytime (local time 10:00 AM, 1:00 PM, and 4:00 PM) in June and December, to represent the aerosol related variability

in the OCO-2 spectral measurements (Rienecker et al., 2011). MERRA aerosol data consisting of five composite types, namely dust (DU), sea salt (SS), sulfate (SU), black carbon (BC), and organic carbon (OC), have nearly zero bias and a correlation coefficient of ~0.9 with respect to the collocated Aerosol Optical Depth (AOD) measurements from AErosol RObotic NETwork (AERONET), Multi-angle Imaging

SpectroRadiometer (MISR), and Ozone Monitoring Instrument (OMI) (Buchard et al., 2015). At each sounding location, the two composite types most common at that location are included in the state vector for the operational retrieval, along with liquid water and ice cloud, and are retrieved by the L2 algorithm. For the analysis presented here we take into account the variability of all five type of aerosols, including those

not retrieved, as described next.

The L2 calculations for linear error analysis are performed at each sounding with the operational state vector and *a priori* uncertainties, augmented as follows. Ten additional aerosol quantities are added to the state vector, namely the AOD for each

of the 5 composite MERRA aerosols, integrated over two layers. Using the relative

pressure scale σ, defined as the fraction of surface pressure, the lower layer is at σ = 0.95 with width 0.05, while the upper layer is at σ = 0.5 with width 0.2. The a priori amount and uncertainty for each of these 10 aerosol quantities is set equal to a small positive number, non-zero to avoid singularity, but small enough to have negligible

effect on the algorithm. The L2 algorithm then calculates the Jacobians for each of these 10 interfering aerosol species.

Subsequently, the linear error analysis combines the Jacobians for all of the aerosol and cloud quantities (liquid water, ice, the 2 types retrieved, and the 10 additional

interfering aerosols) with estimates of the ensemble variability of their total atmospheric AOD, to calculate the resulting error in $XCO_2$. For this step, we have created a database of the standard deviation of each of the 5 MERRA composite types, in 2 layers defined as the surface to 750 hPa and 750 hPa to top of atmosphere, on a 2.5 x 2.5 degree lat/lon grid, for each month. For each sounding location, our

error analysis algorithm looks up the standard deviation at the nearest grid point for all 10 aerosols, and uses that as the estimated ensemble variability. For liquid water and ice cloud, we assume the ensemble variance equals the *a priori* variance. The *a priori* uncertainty of liquid water and ice (approximately a factor 6 (1σ)) was deliberately set large enough to minimize its effect on retrieved $XCO_2$.


The two retrieved aerosol types are counted twice by this procedure, once in the operational state vector and again in the part of the state vector as augmented for the error analysis. To avoid an error due to 'double counting', we set the ensemble variance for the aerosols in the operational state vector to very small values, ensuring

they produce negligible error in retrieved $XCO_2$.

### 3.3.2 Empirical Orthogonal Functions (EOFs)

Interference errors due to the scale factors applied to the operational EOFs are calculated as part of the error analysis by including the actual EOF scale factors in the

state vector. The results show negligible effects on $XCO_2$ uncertainties and degrees of freedom due to these scale factors.



### 3.3.3 Other Interference Errors

Other non-$CO_2$ components of the state vector include surface pressure, water vapor
column, an offset to the a priori temperature, a linear dispersion coefficient for each
spectral band (defining the separation in wavelength between adjacent pixels), albedo,
and the linear change in albedo across each spectral band. Land (nadir) observations
also include a coefficient of fluorescence, and ocean (glint) observations include wind

speed. These have all been included in the error analysis.

For all of these components, an effort has been made to include an estimate of global
variability as the *a priori* uncertainty, and this has been used as the estimated
ensemble variability in the error analysis. The net effect of these uncertainties is fairly
small compared to aerosols and forward model errors, so refining this ensemble

estimate has not been a high priority, but may be considered later.

### 3.4 Forward Model Error

Forward model errors which have been evaluated in this analysis include those due to

a variety of spectroscopic and calibration parameters.

Table 1 shows the estimated uncertainties in spectroscopic parameters used in the L2
algorithm. The parameters listed are those required for the spectroscopic line shape
models used within the OCO-2 v7 L2 algorithm. For $CO_2$ this is a speed-dependent

Voigt line shape with tridiagonal line mixing and for $O_2$, this is a Voigt line shape
with first order line mixing, with a contribution from collision-induced absorption
(CIA). The relevant references, describing these parameters and the uncertainty
estimates, are given in the Table.

The majority of the uncertainties listed in Table 1 are based on published values. The
notable exceptions are speed dependence in the $CO_2$ bands, and line mixing in the $O_2$
A-band. Fairly large uncertainties have been estimated for these by L. Brown at JPL
(private communication 2014).





It is also worth noting that the exponent of the temperature dependence of the pressure broadened linewidths in the $O_2$-A band has been measured recently by Droiun et al (2015). The absolute value of this parameter differs by ~8% from the previously published value (Brown & Plymate, 2000) which was used in the OCO-2 data processing up to and including v7. The newer, Drouin et al value will change the

derived $XCO_2$ values by ~ 1 ppm.

Also of note is a discrepancy between recent measurements of the line strength in the WCO2 band. The values used by the OCO-2 algorithm are based on Devi et al (2007, 2016). Values from Polyansky et al (2015) differ from those in Devi et al (2016) by

~1.2%.

Uncertainties in the calibration parameters are shown in Table 2. These are based on pre-flight laboratory calibration of the instrument at the Jet Propulsion Laboratory. The parameters are defined as follows. The instrument line shape (ILS) in each band

is assumed to have a single uncertainty, in its width. Its shape as measured in the laboratory before launch is assumed to be correct. Radiometric gain is the factor applied to the measured voltages to convert them to absolute physical units. Finally, OCO-2 is only sensitive to one polarization of the incoming radiation, whose angle of orientation is the 'polarization angle'.


In applying the uncertainties in polarization angle, we note that the observed spectrum **S** may be written in terms of the Stokes parameters **I**, **Q**, **U**, and **V**, and Mueller matrix coefficients m_I, m_Q, m_U, and m_V:

$\qquad$ **S** = m_I* **I** + m_Q* **Q** + m_U* **U** + m_V* **V** $\qquad$ (11)

Uncertainties in the Mueller matrix coefficients were calculated as follows:

First,

$$m\_I = 0.5$$
$$m\_Q = 0.5 * \cos(2*\varphi_{pol})$$
$$m\_U = 0.5 * \sin(2*\varphi_{pol})$$
$$m\_V = 0$$





The uncertainty in the polarization angle $\varphi_{pol}$ is ± 0.5° (Table 2, 1σ) for all 3 bands,
m_Q and m_U are derived from the same measurement, so have correlation = 1, and
the 3 bands should be independent. From the above, a 3 x 3 covariance matrix can
readily be calculated which applies to all 3 bands (uncertainty in m_I is assumed to be
non-zero, but very small, to avoid singularity). Note that **V**, the circular component of
polarizaton, is completely ignored in the L2 algorithm as there are very few natural
sources.

## 4. Results

Figures 1 through 6, and Tables 3 and 4, below, display the summary of results for the
off-line error analysis. The data are gridded into 10 by 10 degree bins and only bins
with a minimum of 3 soundings are displayed. An overall observation is that there is
some spatial seasonal dependence in all of the error types due to the shifting sub solar
point of the sun from summer to winter which drives signal and air mass related
errors.

Figure 1 shows measurement error, due to random noise in the measured spectra. It is
typically ~0.5 ppm for a single sounding, and is expected to decrease with averaging
approximately as expected for random error, i.e. in proportion to √N, for N = number
of soundings in the average. The error is smaller and more uniform for ocean than
land, presumably due to the increased SNR in glint viewing mode.

Forward model error, divided into spectroscopic and instrument error, is shown in
Figs. 2 and 3, respectively. Spectroscopic and instrument error make roughly equal
contributions to forward model error. Spectroscopic error in ocean glint observations
shows little variation, and is ~ 1.3 ppm. For land nadir it is more variable, typically 1-
2 ppm. The most important spectroscopic error is due to uncertainty in the WCO$_2$
band strength (Tables 3, 4). Instrument error is somewhat more variable, especially
over land. It is ~ 1 ppm in ocean glint and ~0.5-2.5 in land nadir. The most important
instrument error is due to uncertainty in the instrument line shape (ILS).



Maps of aerosol error are shown in Fig. 4a, and for comparison, the monthly mean aerosol optical depth from MERRA is shown in Fig 4b. In most places, aerosol errors are surprisingly small, typically ~0.5 ppm. However there are regions where they are systematically larger, ~ 2.0 - 2.5 ppm. These regions include east Asia, which has

highly variable aerosol loading, and the tropical North Atlantic, due to dust (not shown), presumably from North Africa. There are also systematically large errors over the Arctic Ocean. We believe these occur because of high sensitivity of the algorithm to small spectral errors at high solar zenith angle.

Variable error is shown in Fig. 5. Comparison to Figs. 3 and 4 shows that variable error over land is dominated by instrument error (due to instrument line shape), but by aerosol error over ocean. It is typically ~0.5 – 2.0 ppm in land nadir, and mostly < 1 ppm in ocean glint, but as for aerosol, it is 2.0 - 2.5 ppm in glint in some regions.

Total error from all sources is shown in Fig. 6. It is ~1.5 - 3.5 ppm over land and ~1.5 - 2.5 ppm over ocean.

**5. Discussion**

Inspection of the global mean and standard deviations in Tables 3 and 4 gives rise to some interesting observations. In general, the fixed, or approximately fixed, error sources (spectroscopy and instrument calibration), cause mean errors much larger

than their standard deviations. This implies that whatever the true value of the error in the relevant forward model parameter, its main effect can in principle be removed by simple bias correction based on validation measurements. Similarly, the error in the difference in $XCO_2$ between two soundings is better characterized by the variability of that parameter's effect than by its mean; i.e the mean effect is the same for both

soundings and is removed by taking their difference. This is the rationale for our definition and use of Variable Error, as discussed in Sect. 3. It is also worth noting that both the mean and standard deviation of errors due to fixed sources is larger for land nadir than for ocean glint soundings.

As noted above, spectroscopic error varies little over the ocean, and modestly over
       land. The main sources of this behavior can be traced to $WCO_2$ and $O_2$ line strength,
       which are the largest error sources but are fairly constant in both regimes, and $SCO_2$
       and $O_2$ line mixing, which have highly variable effects over land, and little variation
       over ocean.


       Three components of instrument error were analyzed. Error due to uncertainty in the
       polarization angle $\varphi_{pol}$ is negligible, $< 0.01$ ppm (not shown elsewhere). Uncertainty
       in instrument gain is a significant but fairly small source of $XCO_2$ error, averaging
       0.2-0.3 ppm. The behavior of the instrument error is dominated by the ILS;

uncertainties in $XCO_2$ due to the ILS are the largest single error source over land, with
       variability second only to aerosol ($\sim 1.4 \pm 0.4$ ppm). Pre-flight measurements of the
       ILS were done to high accuracy (Table 2), but the sensitivity of $XCO_2$ to the ILS is
       high. Error due to the ILS is larger and much more variable over land than over ocean.

Uncertainty due to smoothing error is fairly small. It is typically $\leq 0.2$ ppm. The full
       results (not shown) indicate it is rarely if ever larger than 0.4 ppm. The magnitude of
       smoothing error was deliberately minimized by choice of a loose *a priori* constraint
       on the $CO_2$ profile in the L2 algorithm. It is likely to vary systematically with local
       conditions, since it arises in the difference between the actual and *a priori* $CO_2$ profile

shapes.

       Tables 3 and 4 also emphasize that the dominant variable error is due to aerosol.
       Although the absolute size of the aerosol error is fairly small, it varies widely from
       place to place, with a standard deviation up to 195% of its mean value (the coefficient

of variations are 134%, 109%, 195%, and 132% for June nadir-land, Dec nadir-land,
       June glint-water and Dec glint-water, respectively). Furthermore, it will depend on the
       actual atmospheric aerosol distribution, which will vary in a complex fashion with
       space and time. Correlation of the aerosol distribution is likely to be a major source of
       correlation in $XCO_2$ error, which will be difficult to characterize quantitatively.




## 6. Recommendations for Further Research

We envisage a continual ongoing analysis to quantify uncertainties in the OCO-2
        measurements. We believe such quantification is critical for using the data to
        constrain the geophysical carbon cycle. Linear error analysis as presented here will be
        a key part of that effort, and it is important to replicate the analysis when future
        versions of the L2 algorithm are released and mission data are re-processed. The error
analysis should be extended to further examine errors produced by the algorithm
        itself. This would include studying the effect of errors in algorithm inputs such as the
        *a priori* state vector. A more general subject for study is non-linearity of the forward
        and inverse models. Both of these areas are foci of active research. In the particular
        case of nonlinearity, linear error analysis can be supplemented with Monte Carlo
studies. The Monte Carlo approach can interrogate the probability distribution of
        retrieval errors under specified conditions and can characterize correlations between
        multiple error sources, such as interference and nonlinearity, for example. Monte
        Carlo studies require far more computational effort than the linear error analysis, so
        experiments should be designed for a carefully selected subset of conditions.


        Specific recommendations for linear error analysis include the following. Linear error
        analysis, as applied here to simulations, will be used to estimate uncertainties in
        selected sets of actual OCO-2 measurements. This work will have two main goals.
        First, we will analyze sets of OCO-2 measurements which have been used for 'top-
down' error estimates and validation, by comparison to TCCON data and by
        examining observed scatter in uniform, local areas. The volume of OCO-2 data has
        provided a large collection of validation datasets for many regions, spanning all
        seasons. The results of these 'top-down' estimates will be compared to the 'bottom-
        up' estimates of linear error analysis. If these two types of estimates are consistent
they will give us confidence in our overall understanding of measurement uncertainty.
        Any inconsistencies will require further investigation. One possible source of
        inconsistencies, already under investigation as described above, is non-linearity of the
        forward model.

Second, the variability of the bottom-up estimates will be systematically compared to
        the variation in sounding geometry, atmospheric conditions, and surface type. This



will improve insight into the causes of measurement uncertainty, and guide data users in quantitative applications.


Acknowledgements:

We thank the following members of the OCO-2 team for support and helpful discussions: Vijay Natraj, Linda Brown, Brian Drouin, Chris Benner, Malathy Devi, and Annmarie Eldering. Part of the research was carried out at the Jet Propulsion

Laboratory, California Institute of Technology, under a contract with the National Aeronautics and Space Administration. The CSU contribution to this work was supported by JPL subcontract 1439002. The contribution by BC Scientific Consulting was supported by JPL subcontract 1518224.

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






Table 1: Uncertainties in Spectroscopic Parameters Used in the L2 Algorithm

| | Band | Uncertainty | Reference |
|---|---|---|---|
| Line Strength | | | |
| | $SCO_2$ | 0.40% | Joly et al, 2009 |
| | $WCO_2$ | 0.30% | Devi et al 2007, 2016 |
| | $O_2$ - A | 0.40% | Long et al, 2010 |
| Air Broaden. | | | |
| | $SCO_2$ | 0.15% | Joly et al, 2009 |
| | $WCO_2$ | 0.10% | Devi et al 2007 |
| | $O_2$ - A | 0.20% | Robichaud et al 2008 |
| T – Width | | | |
| | $SCO_2$ | 0.45% | Joly et al, 2009 |
| | $WCO_2$ | 0.60% | Devi et al, 2007 |
| | $O_2 – A$ | 1.25% | Drouin, et al., 2016 |
| CIA | | | |
| | $O_2 – A$ | 0.1* | Long et al, 2012 |
| H2O Broaden. | | | |
| | $SCO_2$ | 3% | Sung et al, 2009** |
| | $WCO_2$ | 3% | Sung et al, 2009** |
| Pressure Shift | | | |
| | $SCO_2$ | 2.6% | Joly et al, 2009 |
| | $WCO_2$ | 1.5% | Devi et al 2007 |
| | $O_2$ - A | 2% | Robichaud et al 2008 |
| Line Mixing | | | |
| | $SCO_2$ | 10% | Benner et al, 2011 |
| | $WCO_2$ | 10% | Benner et al, 2011 |
| | $O_2 – A$ | 10% | estimate** |
| Speed Dep | | | |
| | $SCO_2$ | 10% | estimate** |
| | $WCO_2$ | 10% | estimate** |

\* $10^{-7}$ $cm^{-1}$ $amagat^{-2}$
\*\* L. Brown private communication, 2014




Table 2: Uncertainties in Calibration Parameters Used in the L2 Algorithm

|  | Uncertainties | Correlations |
|---|---|---|
| ILS | | |
| $O_2$ - A | 0.25% | 0.7 to $SCO_2$ and $WCO_2$ |
| $WCO_2$ | 0.25% | 0.8 to $SCO_2$ |
| $SCO_2$ | 0.40% | |
| | | |
| Radiometric Gain | | |
| $O_2$ - A | 1.1% | 0.5 to $WCO_2$ and $SCO_2$ |
| $WCO_2$ | 1.5% | |
| $SCO_2$ | 1.6% | |
| | | |
| Polarization Angle | | |
| $O_2$ - A | 0.5° | |
| $WCO_2$ | 0.5° | |
| $SCO_2$ | 0.5° | |












Table 3:
Global Mean Errors in $XCO_2$ and Standard Deviations for Land Nadir Observations.
Also shown is the coefficient of variation (relative standard deviation)

|  | June | | December | |
|---|---|---|---|---|
| Measurement | 0.55 ± 0.12 | 22% | 0.58 ± 0.12 | 21% |
| $SCO_2$ Line Strength | 0.23 ± 0.15 | 65% | 0.28 ± 0.18 | 64% |
| $WCO_2$ Line Strength | 0.94 ± 0.16 | 17% | 0.92 ± 0.21 | 23% |
| $O_2$ Line Strength | 0.55 ± 0.14 | 25% | 0.57 ± 0.09 | 16% |
| $O_2$ Line Width | 0.27 ± 0.08 | 30% | 0.28 ± 0.05 | 18% |
| $O_2$ Width T Dep. | 0.11 ± 0.06 | 55% | 0.19 ± 0.09 | 47% |
| Line Mixing $SCO_2$ | 0.40 ± 0.26 | 65% | 0.52 ± 0.36 | 69% |
| Line Mixing $O_2$ | 0.21 ± 0.17 | 81% | 0.22 ± 0.17 | 77% |
| Speed Dep. $WCO_2$ | 0.32 ± 0.06 | 19% | 0.34 ± 0.10 | 29% |
| Total Spectroscopy | 1.35 ± 0.17 | 13% | 1.43 ± 0.27 | 19% |
| Radiometric Gain | 0.15 ± 0.09 | 60% | 0.16 ± 0.09 | 56% |
| ILS | 1.39 ± 0.44 | 32% | 1.32 ± 0.32 | 24% |
| Total Instrument | 1.40 ± 0.43 | 31% | 1.33 ± 0.32 | 24% |
| Smoothing | 0.15 ± 0.02 | 13% | 0.19 ± 0.04 | 21% |
| Aerosol Interference | 0.47 ± 0.63 | 134% | 0.43 ± 0.47 | 109% |
| Interference w/o aerosol | 0.18 ± 0.11 | 61% | 0.28 ± 0.16 | 57% |
| Variable | 0.93 ± 0.59 | 63% | 0.94 ± 0.44 | 47% |
| Total | 2.16 ± 0.56 | 26% | 2.17 ± 0.42 | 19% |












Table 4:
Global Mean Errors in $XCO_2$ and Standard Deviations for Ocean Glint Observations
Also shown is the coefficient of variation (relative standard deviation)

| | June | | December | |
|---|---|---|---|---|
| Measurement | 0.35 ± 0.10 | 29% | 0.41 ± 0.07 | 17% |
| $SCO_2$ Line Strength | 0.27 ± 0.09 | 33% | 0.20 ± 0.10 | 50% |
| $WCO_2$ Line Strength | 0.86 ± 0.07 | 8% | 0.91 ± 0.09 | 10% |
| $O_2$ Line Strength | 0.74 ± 0.05 | 7% | 0.72 ± 0.03 | 4% |
| $O_2$ Line Width | 0.37 ± 0.03 | 8% | 0.36 ± 0.02 | 6% |
| $O_2$ Width T Dep. | 0.16 ± 0.03 | 19% | 0.22 ± 0.08 | 36% |
| Line Mixing $SCO_2$ | 0.07 ± 0.05 | 71% | 0.10 ± 0.05 | 50% |
| Line Mixing $O_2$ | 0.28 ± 0.09 | 32% | 0.23 ± 0.08 | 35% |
| Speed Dep. $WCO_2$ | 0.31 ± 0.03 | 10% | 0.33 ± 0.03 | 9% |
| Total Spectroscopy | 1.32 ± 0.04 | 3% | 1.35 ± 0.05 | 4% |
| Radiometric Gain | 0.28 ± 0.10 | 36% | 0.27 ± 0.07 | 26% |
| ILS | 0.88 ± 0.12 | 14% | 0.84 ± 0.16 | 19% |
| Total Instrument | 0.93 ± 0.13 | 14% | 0.88 ± 0.15 | 17% |
| Smoothing | 0.15 ± 0.02 | 13% | 0.15 ± 0.02 | 13% |
| Aerosol | 0.37 ± 0.72* | 195% | 0.19 ± 0.25 | 132% |
| Interference w/o aerosol | 0.06 ± 0.06 | 100% | 0.08 ± 0.05 | 63% |
| Variable | 0.62 ± 0.67* | 108% | 0.52 ± 0.23 | 44% |
| Total | 1.77 ± 0.54* | 31% | 1.69 ± 0.18 | 11% |

*driven by Sahara dust and high latitude outliers







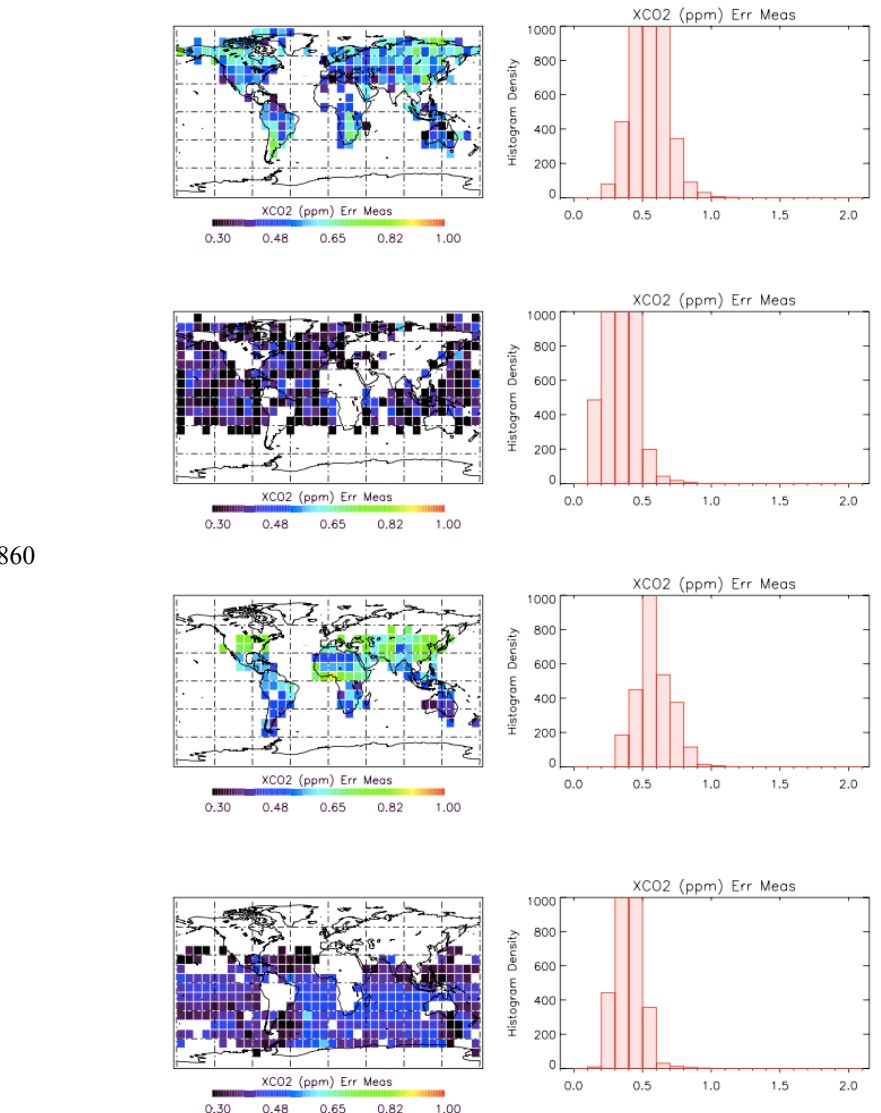


Figure 1 – Measurement Error.

Top: June, land; second: June, ocean; third: Dec. land; bottom: Dec. ocean




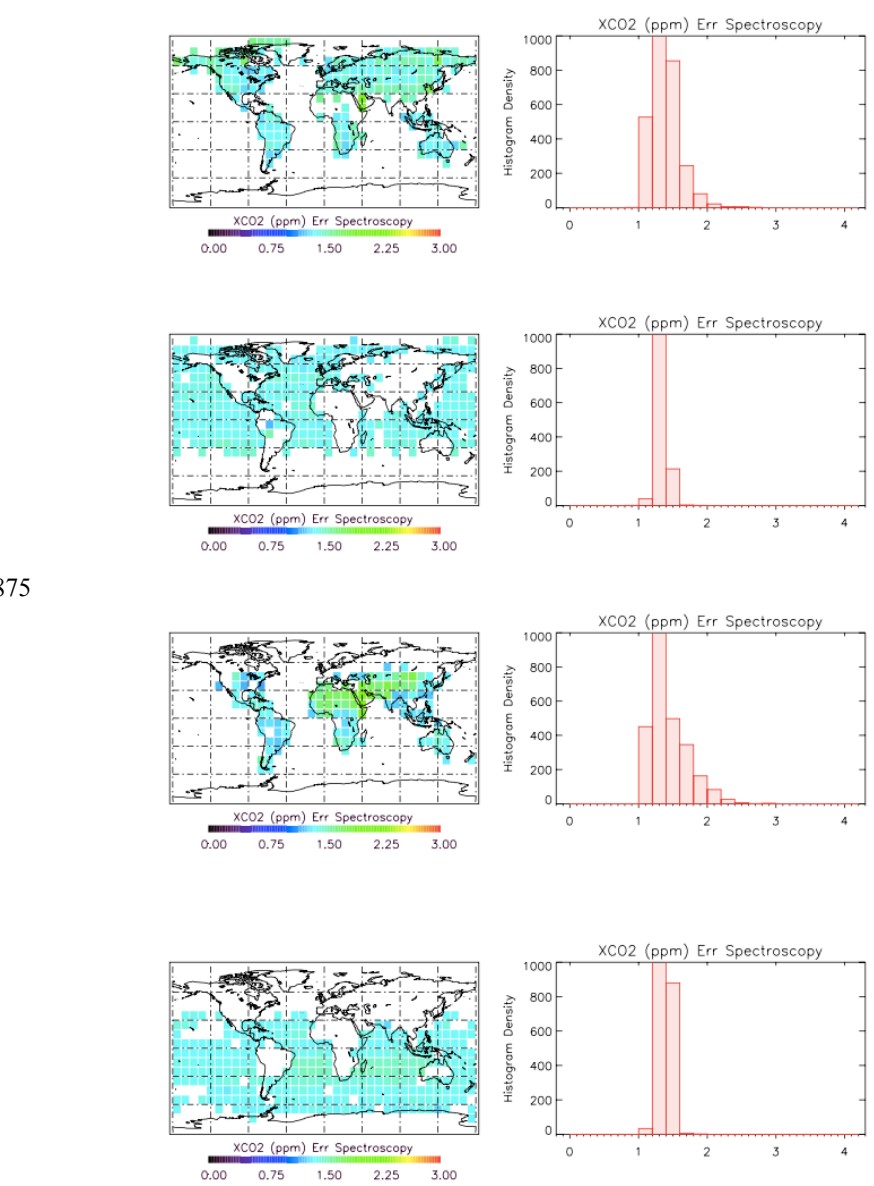


Figure 2 – Error due to spectroscopy

Top: June, land; second: June, ocean; third: Dec. land; bottom: Dec. ocean






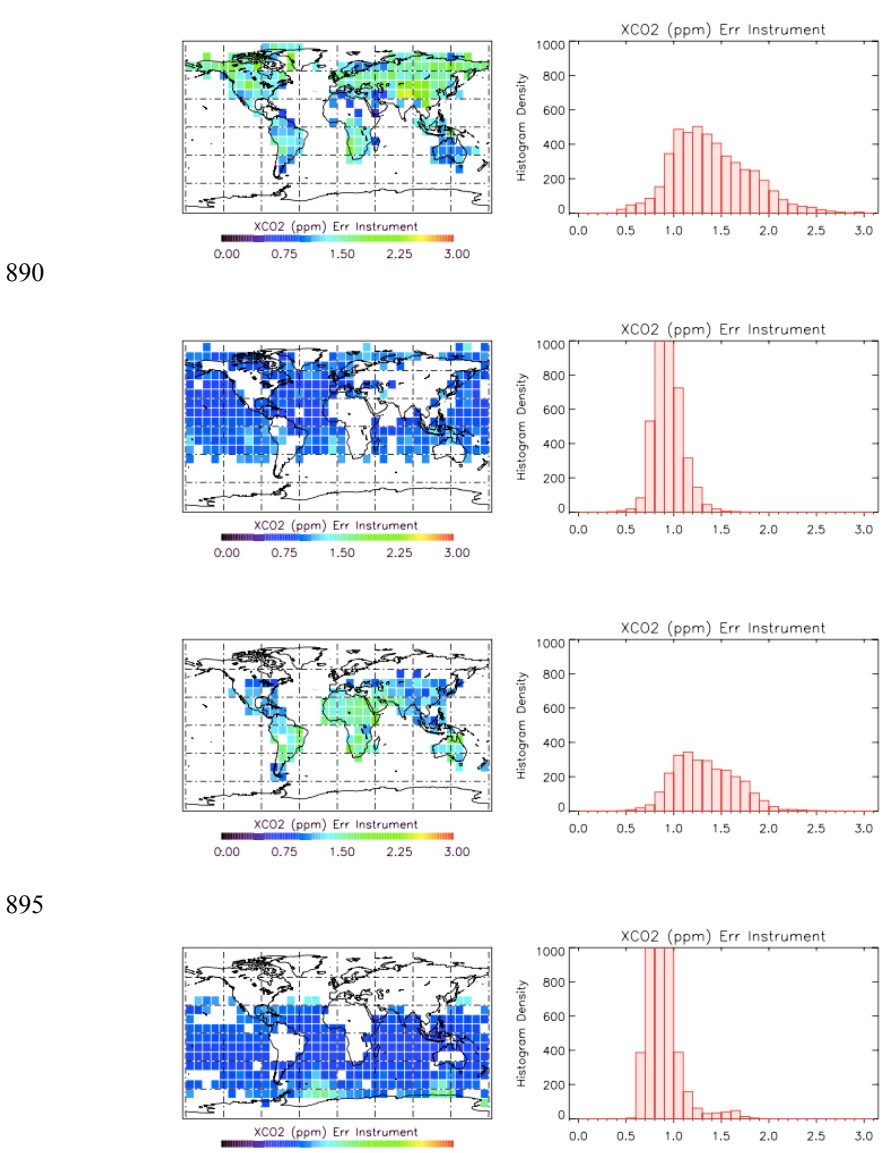



Figure 3 – Instrument Error

900              Top: June, land; second: June, ocean; third: Dec. land; bottom: Dec. ocean





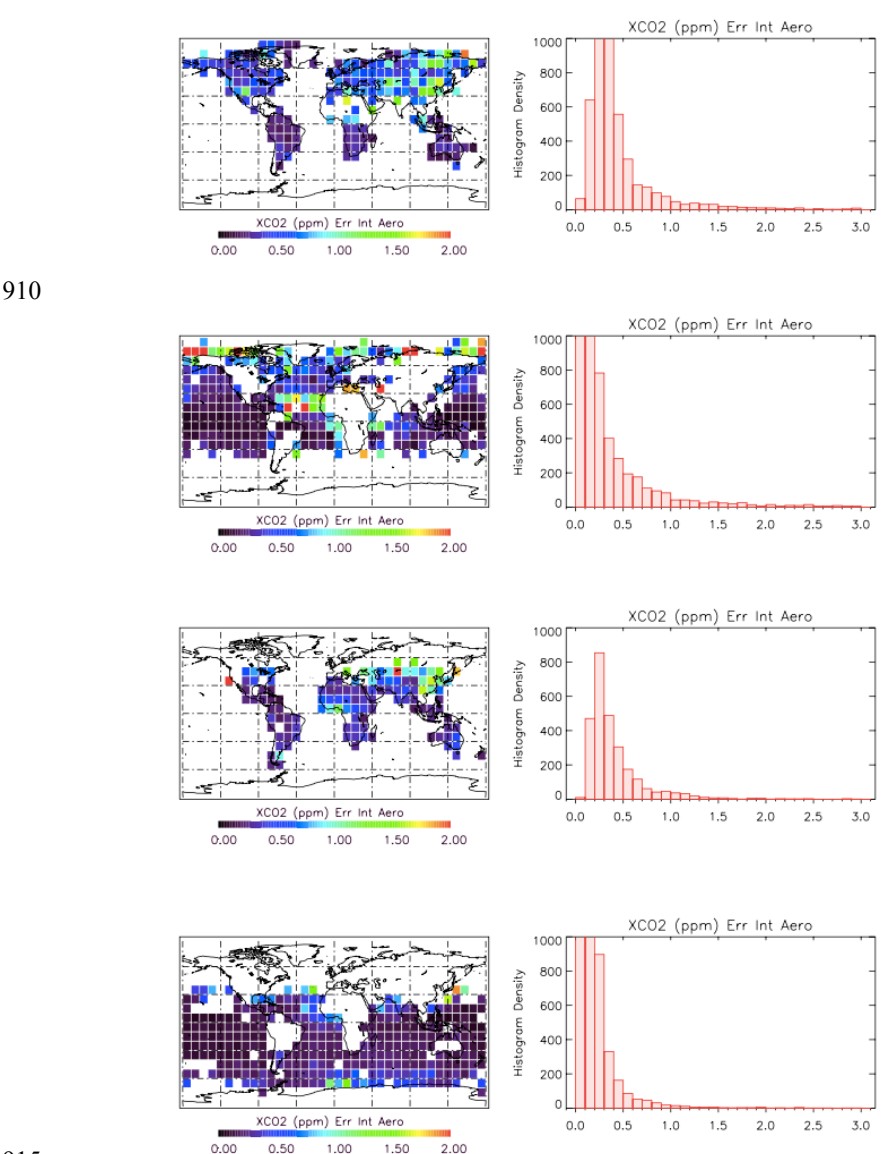


Figure 4a – Aerosol error

Top: June, land; second: June, ocean; third: Dec. land; bottom: Dec. ocean






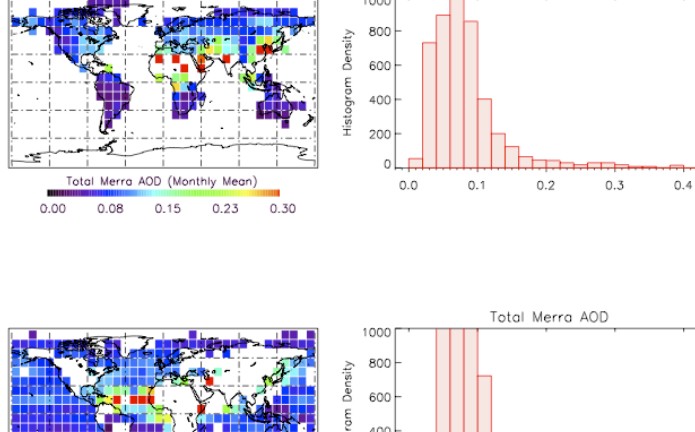


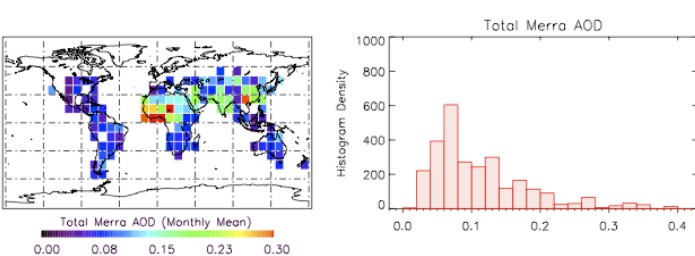

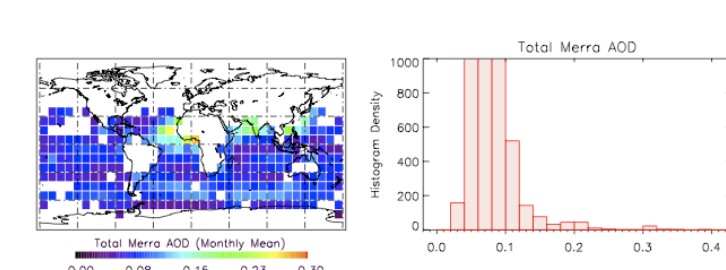


Figure 4b – Monthly Mean Aerosol Optical Depth (AOD) from MERRA

Top: June, land; second: June, ocean; third: Dec. land; bottom: Dec. ocean






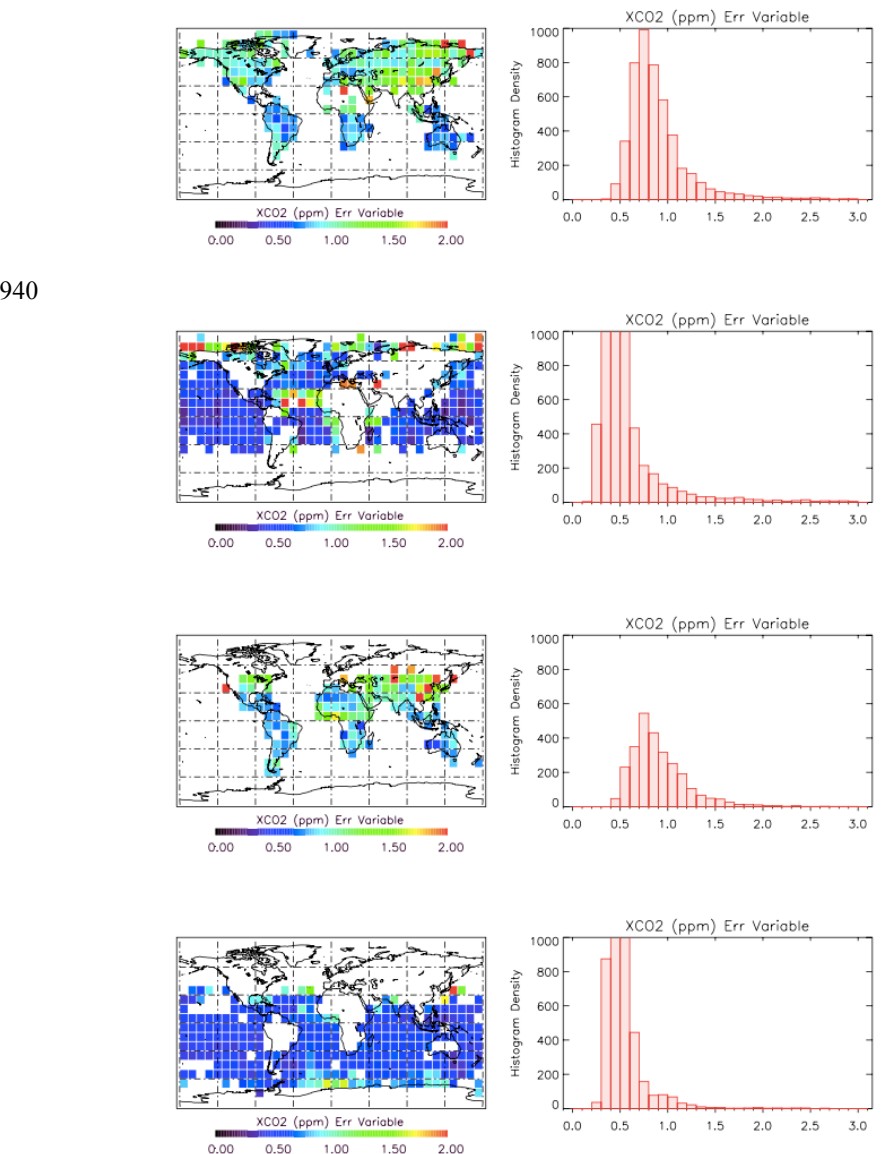


Figure 5 – Variable error

Top: June, land; second: June, ocean; third: Dec. land; bottom: Dec. ocean









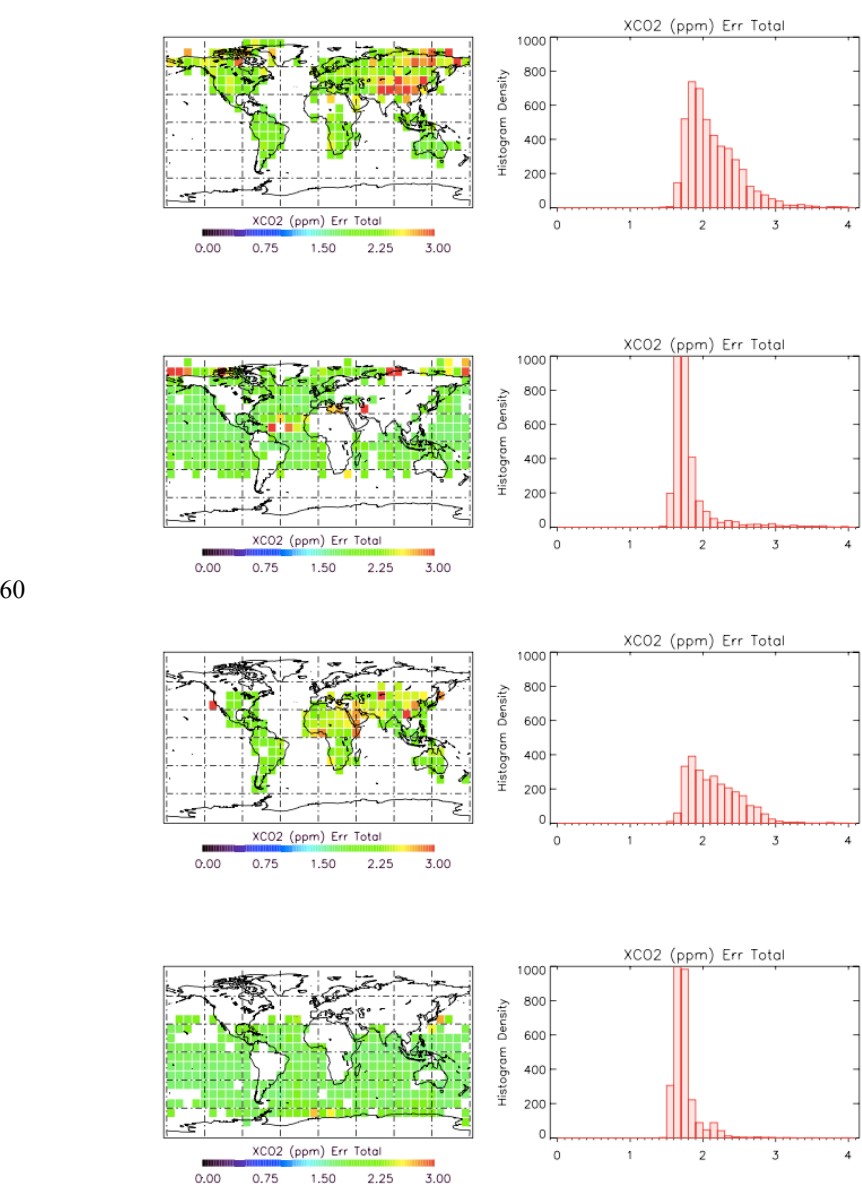

Figure 6 – Total error

Top: June, land; second: June, ocean; third: Dec. land; bottom: Dec. ocean