# Peer review of "Quantification of Uncertainties in OCO-2 Measurements of XCO2: Simulations and Linear Error Analysis"

_Atmospheric Measurement Techniques, 2016_

## Referee Comment (RC1) · Anonymous Referee #1 · 8 Jul 2016

This paper provides a theoretical estimate of OCO measurement error. This instrument has been launched two years ago for the monitoring of CO2 from space. To my knowledge, the analysis of the products shows an excellent random error, but systematic errors (biases) remain and prevent any significant insight on the carbon cycle. In the context, an understanding of the causes for measurement uncertainty is certainly welcome.

The paper is very clearly presented and its structure is appropriate. It is certainly of interest for those who want to understand the origin of measurement errors with OCO or similar remote sensing observations. It can be published as is.

I nevertheless have one simple request : Please provide more information on the

aerosols MERRA product. What is the reference wavelength for the AOD and what is the range of variation of the spectral Angstrom coefficient. Finally, Figure 4b shows many areas with no estimates (white). How are these absence of data dealt with ?

---

## Referee Comment (RC2) · Anonymous Referee #2 · 11 Jul 2016

The paper by Connor et al. uses retrieval simulations and linear error propagation to estimate uncertainties to be attributed to OCO-2 XCO2 measurements. The study builds on the paper by Connor et al., JGR, 2008, but goes substantially beyond by considering OCO's actual instrument performance and by extending the sounding ensemble to a large range of geophysical conditions. The paper contributes new and very relevant aspects, it is well written, methods are robust. I recommend publication after considering my comments below:

1. Why is the "bottom-up" error estimate not compared to the "top-down" error calculated from retrieved and true XCO2 (known from the simulation input)? It would be a first step toward the recommended TCCON comparisons (L519). I would consider

such a comparison essential to confirm the estimated bottom-up sounding errors.

2. Since the paper is a follow-up of Connor et al., JGR, 2008, it might be useful to highlight differences and refinements. In particular, the present study identifies aerosols as a driving error contribution while aerosols were considered less important by the precursor study. I presume the reason is that the present study considers foreign aerosols ie. aerosols which are not part of the retrieval state vector.

3. The definition of “variable error” in contrast to “fixed error sources” could be misleading in a sense that readers could easily conclude that the “fixed error sources” such as spectroscopy and instrument calibration are not really to worry about. In fact, it could be exactly those “fixed error sources” (eg. spectroscopic line shape errors or instrument line shape errors) that induce spurious XCO2 gradients on large spatial and on seasonal scales (eg. through Gy depending on viewing geometry, etc. (L204)). In my opinion, bias correction has a hard time to catch such “slow” variability in contrast to what is written in the manuscript (L455).

Statements such as the following could augment the chance for misunderstanding:

(L29) “we also estimate the ‘variable error’ which differs between soundings, to infer the error in the difference of XCO2 between any two soundings”. Is it really the difference between any two soundings or rather the difference between two spatiotemporally close soundings?

(L 231) “On the other hand, an error which is constant, or at least has a well-defined mean value, can be subtracted from all soundings with minimal or no effect on gradients of XCO2.” While this is certainly true, spectroscopy and instrument calibration are constant error sources, but induce variable XCO2 errors (L203). The induced variability is “slow” (seasonal and continental-scale) but this could be particularly detrimental for carbon cycle insights.

4. What about foreign spectroscopic interference e.g. H2O spectroscopic errors inducing XCO2 errors?

5. Section 3.1.1: As far as I understand, the linear error analysis for aerosols (and clouds) uses the Jacobians at the retrieved state for both, the retrieved and the non-retrieved aerosol parameters. Is this in any way significantly different from and better or worse than using the Jacobians at the true state?

6. Instrument calibration: The correction of radiometric in-orbit degradation might be source of error. Is this source included in the radiometric gain error (<1.6% (table 2))?

7. Check whether the supplemental material is referred to by the manuscript. The error maps in the supplemental material show that, assuming errors of 0.1 AOD (at what wavelength?), XCO2 errors are factors larger than the composite errors shown in Fig. 4a (main manuscript). Does this mean that the ensemble AOD error used for generating Fig. 4a is typically factors less than 0.1 (in particular for the upper layer)? It might be better to plot AOD standard deviation instead of AOD itself in Fig. 4b. Could it be that aerosol variability on MERRA's horizontal resolution is not representative for OCO's ground-pixel size?

8. References: There is a wealth of publications that aim at estimating retrieval errors and retrieval performance from OCO-2, GOSAT, future missions. While I am in favor of conciseness, I find the paper very sparse in citing those.

---

## Author Comment (AC1) · 22 Sep 2016

Response to Anonymous Referee #1

We thank referee #1 for reviewing of our manuscript and for his/her favorable comments and judgment. We will now address the questions raised.

The reference wavelength for MERRA AOD shown is 750 nm, which is very nearly the same as the reference wavelength used in the OCO-2 L2 algorithm (755 nm). In both its forward model and state vector that algorithm utilizes composite aerosol types based on the original MERRA sub-types, as discussed in detail in http://disc.sci.gsfc.nasa.gov/OCO-2/documentation/oco-2-

v6/OCO2_L2_ATBD.V6.pdf, p 28-31 This exact reference will be inserted in the revised text. The Angstrom coefficient is never explicitly calculated but is implicit in these calculations.

The blank areas in Fig. 1-6 represent latitude/longitude cells where fewer than 3 simulated soundings converged in the L2 retrieval and passed the screens described in the first paragraph of p 3 (line 89-93 of the revised manuscript). The MERRA data in Fig 4b are only shown in the corresponding regions.

---

## Author Comment (AC2) · 22 Sep 2016

**Response to Anonymous Referee #2**

We thank referee #2 for a careful review and insightful questions and comments. We address his/her comments sequentially below.

Many comments by both reviewers can only be addressed appropriately with revisions to the text and/or figures in the manuscript. Therefore we have prepared a revised manuscript and will include it by reference below.

Comment #1:
It was never our intention to quantitatively assess the error analysis based on the difference between the 'true' and retrieved results (hereafter R-T). There are 2 reasons for this: first, the main function of such a comparison would be to check the agreement of mathematics and programming of the orbit simulator, retrieval codes, and the error analysis calculations, rather than our understanding of the atmosphere. Second, the R-T results are not directly comparable to the error analysis, in that no forward model errors are present in R-T, because the same spectroscopic and instrument parameters are used in both the orbit simulator and the L2 retrieval code. Further, the ensemble variability of the state vector elements in the real atmosphere must be estimated in the error analysis; the data set of model soundings is too small to reproduce this ensemble variability accurately, and no attempt has been made to do so. Thus we don't believe such a comparison is an important element in presenting these results; nevertheless it is a useful internal 'sanity' check on our calculations, and we have made such comparisons from time to time. We include example figures below, although we do not include them in the paper.

[Figure]

Standard deviation of the R-T values for ocean observations, June. This figure may be approximately compared to the left hand panel in the second row of Fig. 5

Comment #2:
The earlier study (Connor et al, 2008) was primarily concerned with the 'objective, design, and implementation of the OCO inverse method'. It also described the off-line error analysis formulation and code which has been used in the present study, and gave examples, presenting an 'illustration by prospective application to case studies'

There are many quantitative differences between the error estimates of the 2 studies, based on the difference between OCO and OCO-2, and the development of the sophistication of the algorithm (for one thing, its treatment of aerosols) over nearly 10 years (Connor et al 2008 was submitted for publication in 2006). Further, the current paper uses a much more sophisticated treatment of the inputs to the error estimates, including the aerosol ensemble variability, spectroscopic uncertainties, and instrument calibration uncertainties. In hindsight, it is fair to say that the aerosol errors, particularly in the Southern Hemisphere, were underestimated in Connor et al (2008), by a few tenths of a ppm (compare Fig 7 of Connor et al (2008) to Fig. 4a of the current paper)

However we don't believe publishing a quantitative comparison of the 2 papers' error estimates would serve any useful purpose; we have applied the error analysis formulation of Connor et al, (2008), as referenced in the introduction to Sect 2 and 2.1, and believe that is the appropriate relationship between the two works.

Comment #3:
The determination of errors in XCO2 gradients is key to scientific interpretation of the OCO-2 measurements, and the reviewer has been very perceptive to probe further into the nature of 'variable error' and the role of 'fixed error sources' in producing variable error. We will expand and clarify the discussion of these concepts in the paper as follows:

In the abstract we will add the sentence "Spectroscopy and calibration, although they are themselves fixed error sources, also produce important variable errors in $XCO_2$." In section 5 we add "the remaining, variable error, caused by the fixed error sources, is of critical importance."

The reviewer asks whether the "variable error" is "really the difference between any two soundings or rather the difference between two spatiotemporally close soundings?" In section 2.3 we have added this discussion:

"By its nature, variable error has both random components and those which are systematic in the sense that they depend on conditions such as solar zenith angle, atmospheric temperature, pressure, and aerosol, and surface properties. In other words, the concept of variable error applies to the difference between any two soundings, but its magnitude will depend on the difference in conditions between them."

Spatiotemporally close soundings will on average have more similar sounding conditions, and thus less variable error, than more distant soundings.

Comment #4:
This is an important question which we considered carefully in assembling the list of spectroscopic parameters in Table 1, ultimately deciding that no parameters of interfering species need be included. We have now added a short paragraph in section 3.4 discussing this conclusion.

Comment #5:
The gain function, and thus the averaging kernels, are the properties of the retrieval as actually performed, i.e. they depend on the Jacobians at the retrieved state rather than the true state.

Also, the assumption behind use of linear analysis is that the forward model is (approximately) linear within the error bounds of the retrieval; given that is true, the difference between using the true and retrieved states to calculate the Jacobians will be insignificant.

Comment #6:
The correction of in-orbit degradation of the calibration has not been included.

Comment #7:
Fig 4c has been added to the revised manuscript, showing the MERRA AOD variability. The supplementary material has been explicitly referred to in the text after introducing Fig 4.

Comment #8:
The references in sections 1 and 2 have been expanded to include 4 additional papers addressing the GOSAT retrieval algorithm and earlier assessments of error sensitivity for OCO-2 and GOSAT.

**Quantification of Uncertainties in OCO-2 Measurements of XCO₂: Simulations and Linear Error Analysis**

[revised manuscript text omitted]

The paper is organized as follows. In Sect. 2 we briefly discuss the L2 retrieval algorithm and then present details of the error analysis methodology. This is followed by an enumeration and discussion of the error sources to be considered in Sect. 3. Sect. 4 contains the results of the linear error analysis. Sect. 5 is a discussion of the results, and Sect. 6 identifies needs for future research.

**2. Background and Methodology**

The OCO-2 level 2 full physics retrieval algorithm ('L2'), consists of a forward model and inverse method, described in full detail in JPL (2015). The forward model is a radiative transfer model of the atmosphere coupled to a model of the solar spectrum to calculate the monochromatic spectrum at the top of the atmosphere, which is then convolved with the response function as measured for the OCO-2 instrument. The inverse method is a maximum a posteriori likelihood method of a type which has been widely used in the community (Rodgers, 2000; Rodgers & Connor, 2003; Connor et al, 2008, O'Dell et al, 2012). For comparison, the retrieval algorithm for the spectrally similar measurements by the GOSAT satellite is described in Yoshida et al (2011). Uncertainty in the OCO-2 measurements of $XCO_2$ has been assessed using an off-line error analysis code developed for the purpose (Connor et al, 2008).

**2.1 Formulation**

The error analysis algorithm performs a linear analysis using Jacobians calculated by the operational OCO-2 forward model. This section closely follows the discussion in Connor et al, (2008).

As defined in JPL (2015), $S_a$ is the *a priori* covariance matrix, $S_\varepsilon$ is the measurement error covariance matrix, and $K$ is the weighting function (Jacobian) matrix. The off-line calculations are more detailed and more realistic than error estimates performed operationally. For example, if forward model errors are included in the $S_\varepsilon$ matrix used operationally, the retrieved state may be systematically biased by the *a priori* state. Thus we evaluate the effect of forward model errors off-line. Further, evaluation of the smoothing and interference errors strictly requires the covariance of the ensemble of true states, $S_c$, which is not necessarily equal to the *a priori* covariance $S_a$

(Rodgers & Connor, 2003). The authors' experience with other remote sensing retrievals suggests that the *a priori* constraint, embodied in $S_a$, should be as uniform as practical over all soundings, to avoid introducing an additional source of variability. However, the covariance of true states, $S_c$, varies with latitude, longitude, and season. Estimates of $S_c$ are readily included in the off-line error estimates. (See for example section 3.3.1.)

Equations 1 through 6 follow the definitions of Rodgers (2000) and Rodgers & Connor (2003). Given $K$, $S_\varepsilon$, and $S_a$, we first characterize the operational retrieval by calculating the gain function $G_y$ and the averaging kernel matrix $A$:

$$G_y = (K^T S_\varepsilon^{-1} K + S_a^{-1})^{-1} K^T S_\varepsilon^{-1} \tag{1}$$

and

$$A = G_y K \tag{2}$$

We then specify a list of estimated errors to include in the calculation, and where possible the correlation between errors. We will refer to these as 'error sources.' Next we assemble these into an ensemble covariance $S_c$ (for elements in the state vector) and a forward model parameter covariance $S_b$ (for elements not included in the state vector). Finally, we calculate the Jacobian matrix with respect to the forward model parameters, denoted $K_b$.

For each error in the list, we calculate the resulting covariance of the retrieved state vector, as follows. For measurement error,

$$\hat{S}_m = G_y S_\delta G_y^T \tag{3}$$

where $S_\delta$ may be equal to $S_\varepsilon$, or an alternative estimate of the actual measurement covariance. In the work presented here, $S_\delta = S_\varepsilon$.

For forward model error,

$$\hat{S}_f = G_y\, K_b\, S_b\, K_b^T\, G_y^T \qquad (4)$$

For smoothing error,

$$\hat{S}_s = (A - I)\, S_c (A - I)^T \qquad (5)$$

where I is the identity matrix.

And for interference error, which refers to error in $CO_2$ caused by non-$CO_2$ components of the state vector

$$\hat{S}_i = A_{ue}\, S_{ec}\, A_{ue}^T \qquad (6)$$

where $S_{ec}$ is the ensemble covariance for the non $CO_2$ elements $e$, and $A_{ue}$ is the off-diagonal block of the averaging kernel matrix which relates $e$ to the $CO_2$ profile $u$.

Finally, the total covariance is

$$\hat{S} = \hat{S}_m + \hat{S}_s + \hat{S}_i + \hat{S}_f \qquad (7)$$

and the resulting variance of $XCO_2$ is $\sigma^2_{XCO2} = h^T \hat{S}\, h$, where $h = \partial X_{CO_2} / \partial x$ represents the pressure weighting function. Alternatively, one may calculate the variance in $XCO_2$ due to a given error source, r, as $\sigma_r^2 = h^T\, \hat{S}_r\, h$ and sum the variances for all r.

The discussion of the preceding paragraph makes two assumptions. One, that the retrieval is approximately linear within the region bounded by its uncertainty, and two, that the error sources considered are themselves uncorrelated. Whenever error sources are correlated, the correlations must be included in, e.g. Eq (4) or (6), and the net effect on $XCO_2$ calculated for the combined correlated sources.

**2.2 Treatment of Fixed Error Sources**

Many of the error sources we will consider do not vary randomly, and some do not vary at all.  Spectroscopic errors belong to the class of error sources which are truly fixed. Unfortunately, due to the varying amount of information in each measured spectrum relative to the *a priori* constraint, embodied in changes in the gain function, $\mathbf{G_y}$, the resulting errors in retrieved $XCO_2$ are not fixed. We will treat such errors as follows.

We note that the gain function, $\mathbf{G_y}$, represents the sensitivity of the state vector to the measured radiances. Combining that with the definition of $\mathbf{K_b}$, and considering for the moment a single scalar parameter, the error caused by parameter $\mathbf{b}$ is

$$\mathbf{\hat{x} - x = h^T G_y K_b db} \tag{8}$$

where $\mathbf{\hat{x} - x}$ is the retrieved $XCO_2$ minus the true $XCO_2$, or we may write

$$\mathbf{\hat{x} - x = (h^T G_y K_b db^2 K_b^T G_y^T h)^{1/2}} \tag{9}$$

Replacing $\mathbf{db}^2$ with its matrix equivalent $\mathbf{S_b}$, then for an ensemble of retrievals,

$$\sigma_x = \mathbf{rms(\hat{x} - x) = rms[(h^T G_y K_b S_b K_b^T G_y^T h)^{1/2}]} \tag{10}$$

So if $\mathbf{db}$ is a constant, the error $\mathbf{\hat{x} - x}$ caused by it will vary about a mean value given by $\sigma_x$.  While the true error in parameter $\mathbf{b}$ is an unknown constant, we assume that error is equal to the uncertainty in $\mathbf{b}$.

**2.3 Variable Error**

Sources and sinks of $CO_2$ and the circulation of the atmosphere produce temporal and spatial gradients in the $XCO_2$ field, which are quantitatively predicted by carbon cycle models. Measuring these gradients is a strong test of such models. Thus, errors which vary from sounding to sounding limit the efficacy of the OCO-2 measurements in constraining carbon cycle models. On the other hand, an error which is constant, or at least has a well-defined mean value, can be subtracted from all soundings with minimal or no effect on gradients of $XCO_2$. Therefore, we have attempted to distinguish the uncertainty which differs between soundings, i.e. applies to a difference in two soundings, from the total accuracy.

We will refer to this differential uncertainty as 'variable error'. By its nature, variable error has both random components and those which are systematic in the sense that they depend on conditions such as solar zenith angle, atmospheric temperature, pressure, and aerosol, and surface properties. In other words, the concept of variable error applies to the difference between any two soundings, but its magnitude will depend on the difference in conditions between them.

Our quantitative estimate of variable error is a composite error calculated from a selection taken from all error sources described above. Variable error will be calculated from all error sources, but will exclude the mean error produced by fixed error sources as discussed in Sect. 2.2. Then a first approximation to the predicted error in the difference of $XCO_2$ between two soundings will simply be the variable error multiplied by $\sqrt{2}$, assuming remaining errors are uncorrelated in space or time. This should be equivalent to estimating the net uncertainty in each sounding, and assuming a simple bias correction relative to validation observations has been performed.

Brian Connor 22/9/2016 9:35 AM

**3. Error Types**

We will consider four types of error: measurement, smoothing, interference, and forward model.

Brian Connor 22/9/2016 9:35 AM

**3.1 Measurement error**

The first and most obvious error is random noise in the measured spectrum. This is calculated based on the operational noise model (JPL, 2015), and its direct effect on $XCO_2$ is calculated, and tabulated as 'measurement error.'

However, it is observed that spectral residuals do not decrease with averaging as would be expected for pure random noise, but instead have a systematic structure. Because of this it was decided to derive empirical orthogonal functions (EOFs) representing this systematic structure, and to retrieve scale factors for these functions at every sounding. (See Section 3.3.2.6 of JPL, (2015)). Uncertainties in this process are to be addressed as interference error, below.

**3.2 Smoothing error**

This represents error due to the *a priori* constraint of the state vector. As suggested by Rodgers & Connor (2003), we have separated this into two components. The first, smoothing by the true $CO_2$ profile, which we simply refer to as 'smoothing', is discussed here. The second component is error introduced into $XCO_2$ by the non-$CO_2$ elements of the state vector, which we call 'interference', discussed in the following section.

The error due to the true atmospheric $CO_2$ profile would be best estimated by using the covariance of the ensemble of true states, $S_c$. Exactly which states to include in the ensemble is not well defined. We have chosen to use the *a priori* covariance $S_a$, which is intended to represent the variability of $CO_2$ globally throughout the year. We will systematically overestimate the smoothing error as a result. However, the smoothing error is always small, as we will see, and the use of $S_a$ is fundamentally conservative.

**3.3 Interference error**

3.3.1 Aerosol and Cloud

We apply the Modern Era Retrospective analysis for Research and Applications (MERRA) aerosol reanalysis climatology for daytime (local time 10:00 AM, 1:00 PM, and 4:00 PM) in June and December, to represent the aerosol related variability in the OCO-2 spectral measurements (Rienecker et al., 2011). The MERRA aerosol
data is the basis for the OCO-2 forward model's aerosol types, as described in detail in JPL, 2015, p 28-31. MERRA aerosol data consisting of five composite types, namely dust (DU), sea salt (SS), sulfate (SU), black carbon (BC), and organic carbon (OC), have nearly zero bias and a correlation coefficient of ~0.9 with respect to the collocated Aerosol Optical Depth (AOD) measurements from AErosol RObotic
NETwork (AERONET), Multi-angle Imaging SpectroRadiometer (MISR), and Ozone Monitoring Instrument (OMI) (Buchard et al., 2015). At each sounding location, the two composite types most common at that location are included in the state vector for the operational retrieval, along with liquid water and ice cloud, and are retrieved by the L2 algorithm. For the analysis presented here we take into account the variability
of all five type of aerosols, including those not retrieved, as described next.

The L2 calculations for linear error analysis are performed at each sounding with the operational state vector and *a priori* uncertainties, augmented as follows. Ten additional aerosol quantities are added to the state vector, namely the AOD for each
of the 5 composite MERRA aerosols, integrated over two layers. Using the relative pressure scale $\sigma$, defined as the fraction of surface pressure, the lower layer is at $\sigma = 0.95$ with width 0.05, while the upper layer is at $\sigma = 0.5$ with width 0.2. The a priori amount and uncertainty for each of these 10 aerosol quantities is set equal to a small positive number, non-zero to avoid singularity, but small enough to have negligible
effect on the algorithm. The L2 algorithm then calculates the Jacobians for each of these 10 interfering aerosol species.

Subsequently, the linear error analysis combines the Jacobians for all of the aerosol and cloud quantities (liquid water, ice, the 2 types retrieved, and the 10 additional interfering aerosols) with estimates of the ensemble variability of their total atmospheric AOD, to calculate the resulting error in $XCO_2$. For this step, we have created a database of the standard deviation of each of the 5 MERRA composite types, in 2 layers defined as the surface to 750 hPa and 750 hPa to top of atmosphere, on a 2.5 x 2.5 degree lat/lon grid, for each month. For each sounding location, our error analysis algorithm looks up the standard deviation at the nearest grid point for all 10 aerosols, and uses that as the estimated ensemble variability. For liquid water and ice cloud, we assume the ensemble variance equals the *a priori* variance. The *a priori* uncertainty of liquid water and ice (approximately a factor 6 ($1\sigma$)) was deliberately set large enough to minimize its effect on retrieved $XCO_2$.

The two retrieved aerosol types are counted twice by this procedure, once in the operational state vector and again in the part of the state vector as augmented for the error analysis. To avoid an error due to 'double counting', we set the ensemble variance for the aerosols in the operational state vector to very small values, ensuring they produce negligible error in retrieved $XCO_2$.

**3.3.2 Empirical Orthogonal Functions (EOFs)**

Interference errors due to the scale factors applied to the operational EOFs are calculated as part of the error analysis by including the actual EOF scale factors in the state vector. The results show negligible effects on $XCO_2$ uncertainties and degrees of freedom due to these scale factors.

**3.3.3 Other Interference Errors**

Other non-$CO_2$ components of the state vector include surface pressure, water vapor column, an offset to the a priori temperature, a linear dispersion coefficient for each spectral band (defining the separation in wavelength between adjacent pixels), albedo, and the linear change in albedo across each spectral band. Land (nadir) observations also include a coefficient of fluorescence, and ocean (glint) observations include wind speed. These have all been included in the error analysis.

For all of these components, an effort has been made to include an estimate of global variability as the *a priori* uncertainty, and this has been used as the estimated ensemble variability in the error analysis. The net effect of these uncertainties is fairly small compared to aerosols and forward model errors, so refining this ensemble estimate has not been a high priority, but may be considered later.

**3.4 Forward Model Error**

Forward model errors which have been evaluated in this analysis include those due to a variety of spectroscopic and calibration parameters.

Table 1 shows the estimated uncertainties in spectroscopic parameters used in the L2 algorithm. The parameters listed are those required for the spectroscopic line shape models used within the OCO-2 v7 L2 algorithm. For $CO_2$ this is a speed-dependent

Voigt line shape with tridiagonal line mixing and for $O_2$, this is a Voigt line shape with first order line mixing, with a contribution from collision-induced absorption (CIA). The relevant references, describing these parameters and the uncertainty estimates, are given in the Table.

The majority of the uncertainties listed in Table 1 are based on published values. The notable exceptions are speed dependence in the $CO_2$ bands, and line mixing in the $O_2$ A-band. Fairly large uncertainties have been estimated for these by L. Brown at JPL (private communication 2014).

It is also worth noting that the exponent of the temperature dependence of the pressure broadened linewidths in the $O_2$-A band has been measured recently by Droiun et al (2015). The absolute value of this parameter differs by ~8% from the previously published value (Brown & Plymate, 2000) which was used in the OCO-2 data processing up to and including v7. The newer, Drouin et al value will change the derived $XCO_2$ values by ~ 1 ppm.

Also of note is a discrepancy between recent measurements of the line strength in the WCO2 band. The values used by the OCO-2 algorithm are based on Devi et al (2007, 2016). Values from Polyansky et al (2015) differ from those in Devi et al (2016) by ~1.2%.

Spectroscopic uncertainties in interfering gas species are not a significant source of error in retrieved $XCO_2$. The strongest interferent, by far, is $H_2O$, and its largest uncertainties are in its pressure broadening. Earlier tests of the L2 algorithm (not part of the present analysis) evaluated $H_2O$ line parameters with broadening coefficients 20% different from reference values, and found mean $XCO_2$ changes of $< 0.01$ ppm, with apparently random distribution. Therefore we have not included spectroscopic uncertainties in any interfering species in Table 1, and will not discuss them further.

Uncertainties in the calibration parameters are shown in Table 2. These are based on pre-flight laboratory calibration of the instrument at the Jet Propulsion Laboratory. The parameters are defined as follows. The instrument line shape (ILS) in each band is assumed to have a single uncertainty, in its width. Its shape as measured in the laboratory before launch is assumed to be correct. Radiometric gain is the factor applied to the measured voltages to convert them to absolute physical units. Finally, OCO-2 is only sensitive to one polarization of the incoming radiation, whose angle of orientation is the 'polarization angle'.

In applying the uncertainties in polarization angle, we note that the observed spectrum **S** may be written in terms of the Stokes parameters **I**, **Q**, **U**, and **V**, and Mueller matrix coefficients m_I, m_Q, m_U, and m_V:

$$\mathbf{S} = m\_I* \mathbf{I} + m\_Q*\mathbf{Q} + m\_U*\mathbf{U} + m\_V*\mathbf{V} \tag{11}$$

Uncertainties in the Mueller matrix coefficients were calculated as follows: First,

$$
\begin{aligned}
m\_I &= 0.5 \\
m\_Q &= 0.5 * \cos(2*\varphi_{pol}) \\
m\_U &= 0.5 * \sin(2*\varphi_{pol}) \\
m\_V &= 0
\end{aligned}
$$

The uncertainty in the polarization angle $\phi_{pol}$ is $\pm 0.5°$ (Table 2, $1\sigma$) for all 3 bands, m_Q and m_U are derived from the same measurement, so have correlation = 1, and the 3 bands should be independent. From the above, a 3 x 3 covariance matrix can readily be calculated which applies to all 3 bands (uncertainty in m_I is assumed to be non-zero, but very small, to avoid singularity). Note that **V**, the circular component of polarizaton, is completely ignored in the L2 algorithm as there are very few natural sources.

**4. Results**

Figures 1 through 6, and Tables 3 and 4, below, display the summary of results for the off-line error analysis. The data are gridded into 10 by 10 degree bins and only bins with a minimum of 3 soundings are displayed. An overall observation is that there is some spatial seasonal dependence in all of the error types due to the shifting sub solar point of the sun from summer to winter which drives signal and air mass related errors.

Figure 1 shows measurement error, due to random noise in the measured spectra. It is typically ~0.5 ppm for a single sounding, and is expected to decrease with averaging approximately as expected for random error, i.e. in proportion to $\sqrt{N}$, for N = number of soundings in the average. The error is smaller and more uniform for ocean than land, presumably due to the increased SNR in glint viewing mode.

Forward model error, divided into spectroscopic and instrument error, is shown in Figs. 2 and 3, respectively. Spectroscopic and instrument error make roughly equal contributions to forward model error. Spectroscopic error in ocean glint observations shows little variation, and is ~ 1.3 ppm. For land nadir it is more variable, typically 1-2 ppm. The most important spectroscopic error is due to uncertainty in the $WCO_2$ band strength (Tables 3, 4). Instrument error is somewhat more variable, especially over land. It is ~ 1 ppm in ocean glint and ~0.5-2.5 in land nadir. The most important instrument error is due to uncertainty in the instrument line shape (ILS).

Maps of aerosol error are shown in Fig. 4a, and for comparison, the monthly mean aerosol optical depth from MERRA is shown in Fig 4b and its standard deviation in Fig 4c. The sensitivity of $XCO_2$ to interference error caused by the various aerosol types is shown in the Supplementary Material published with this paper.

[revised manuscript text omitted]

Boesch, H., D. Baker, B. Connor, D. Crisp and C. Miller, (2011). Global Characterization of $CO_2$ Column Retrievals from Shortwave-Infrared Satellite Observations of the Orbiting Carbon Observatory-2 Mission, *Remote Sens.,* 3(2), 270-304.

Buchard V., da Silva A. M., Colarco P. R., Darmenov A., Randles C. A., Govindaraju R., Torres O., Campbell J., and Spurr R. (2015). Using the OMI aerosol index and absorption aerosol optical depth to evaluate the NASA MERRA Aerosol Reanalysis, *Atmos. Chem. Phys.*, 15, 5743-5760.

Butz A., Hasekamp, O.P., Frankenberg, C., Aben, I.  (2009). Retrievals of atmospheric CO2 from simulated space-borne measurements of backscattered near-infrared sunlight: accounting for aerosol effects, *Appl. Opt.* 48, No. 18.

Connor, B. J., Bösch, H., Toon, G., Sen, B., Miller, C., and Crisp, D. (2008), Orbiting Carbon Observatory: Inverse method and prospective error analysis, *J. Geophys. Res.,* 113, D05305, doi:10.1029/2006JD008336.

Crisp, D., Miller, C., and DeCola, P. (2008), NASA Orbiting Carbon Observatory;
measuring the columnaveraged carbon dioxide mole fraction from space, *J. Appl. Remote Sens.*, 2, 023508,  doi:10.1117/1.2898457.

Brian Connor 22/9/2016 9:35 AM

Devi, V. M., Benner, D.C., Brown, L.R., Miller, C.E., and Toth, R.A. (2007), Line mixing and speed dependence in $CO_2$ at 6227.9 cm-1: Constrained multispectrum analysis of intensities and line shapes in the 30013 ← 00001 band, *J. Mol. Spectrosc*. 245, 52-80.

Devi, V.M., Benner, D.C., Sung, K., Brown, L.R., Crawford, T.J., Miller, C.E., Drouin, B.J., Payne, V.H., Yu, S., Smith, M.A.H., Mantz, A.M., and Gamache, R.R. (2016), Line parameters including temperature dependences of self- and air-broadened line shapes of $^{12}C^{16}O_2$: 1.6-μm region, *J. of Quant. Spectrosc. and Rad. Transfer*, vol. 177, 117-144.

Drouin, B.J., Benner, D.C., Brown, L.R., Cich, M.J., Crawford, T.J., Devi, V.M., Guillaume, A., Hodges, J.T., Mlawer, E.J., Robichaud, D.J., Oyafuso, F., Payne, V.H., Sung, K., Wishnow, E.H., and Yu, S. (2016). Multispectrum analysis of the oxygen A-band, *J. of Quant. Spectrosc. and Rad. Transfer*, in press.

Joly, L., Marnas, F., Gibert, F., Bruneau, D., Grouiz, B., Flamant, P.H., Durry, G., Dumelie, N., Parvitte, B., and Zeninari, V., (2009). Laser diode absorption spectroscopy for accurate $CO_2$ line parameters at 2 microns: consequences for space-based DIAL measurements and potential biases, *Applied Optics*, vol. 48 (29), 5475-5483

JPL, OCO-2 Level 2 Full Physics Retrieval Algorithm Theoretical Basis, Version 2.0 Rev 2, March 13, 2015: http://disc.sci.gsfc.nasa.gov/OCO-2/documentation/oco-2-v6/OCO2_L2_ATBD.V6.pdf

Jung, Y., Kim, J., Kim, W. Boesch, H., Lee, H., Cho. C. Tae-Young, G. (2016). Impact of Aerosol Property on the Accuracy of a $CO_2$ Retrieval Algorithm from Satellite Remote Sensing, *Remote Sens*. 8(4), 322. doi:10.3390/rs8040322

[revised manuscript text omitted]

*driven by Sahara dust and high latitude outliers

[Figure]

Figure 1 – Measurement Error.

Top: June, land; second: June, ocean; third: Dec. land; bottom: Dec. ocean

[Figure]

Figure 2 – Error due to spectroscopy

Top: June, land; second: June, ocean; third: Dec. land; bottom: Dec. ocean

[Figure]

Figure 3 – Instrument Error

Top: June, land; second: June, ocean; third: Dec. land; bottom: Dec. ocean

[Figure]

Figure 4a – Aerosol error

Top: June, land; second: June, ocean; third: Dec. land; bottom: Dec. ocean

[Figure]

Figure 4b – Monthly Mean Aerosol Optical Depth (AOD) from MERRA

Top: June, land; second: June, ocean; third: Dec. land; bottom: Dec. ocean

Brian Connor 22/9/2016 9:35 AM

[Figure]

[Figure]

[Figure]

[Figure]

Figure 4c – Monthly Standard Deviation of Aerosol Optical Depth (AOD) from MERRA

Top: June, land; second: June, ocean; third: Dec. land; bottom: Dec. ocean

[Figure]

Figure 5 – Variable error

Top: June, land; second: June, ocean; third: Dec. land; bottom: Dec. ocean

[Figure]

Figure 6 – Total error

Top: June, land; second: June, ocean; third: Dec. land; bottom: Dec. ocean